DATA RELEASE

# Genome assembly of the numbat (*Myrmecobius fasciatus*), the only termitivorous marsupial

Emma Peel[1], Luke Silver[1], Parice Brandies[1], Takashi Hayakawa[2,3], Katherine Belov[1] and Carolyn J. Hogg[1,*]

1 School of Life and Environmental Sciences, The University of Sydney, Sydney, New South Wales, Australia
2 Faculty of Environmental Earth Science, Hokkaido University, Sapporo, Hokkaido, Japan
3 Japan Monkey Centre, Inuyama, Aichi, Japan

## ABSTRACT

The numbat (*Myrmecobius fasciatus*) is an endangered Australian marsupial, and the last surviving member of the Myrmecobiidae family. The numbat regularly undergoes torpor and is unique amongst marsupials as it is the only diurnal and termitivorous species. Here we sequenced the first draft genome of the numbat using 10× Genomics Chromium linked-read technology, resulting in a 3.42 Gbp genome with a scaffold N50 of 223 kbp. A global transcriptome from liver, lung and tongue was also generated to aid genome annotation, identifying 21,465 protein-coding genes. To investigate adaptation to the numbat's termitivorous diet and arid/semi-arid range, we interrogated the most highly expressed transcripts within the tongue and manually annotated taste, vomeronasal and aquaporin gene families. Antimicrobial proteins and proteins involved in digestion were highly expressed in the tongue, alongside umami taste receptors. However, sweet taste receptors were not expressed in this tissue, which combined with the putative contraction of the bitter taste receptor gene repertoire in the numbat genome, may indicate a potential evolutionary adaptation to their specialised termitivorous diet. Vomeronasal and aquaporin gene repertoires were similar to other marsupials. The draft numbat genome is a valuable tool for conservation and can be applied to population genetics/genomics studies and to investigate the unique biology of this interesting species.

Submitted: 17 December 2021

* Corresponding author. E-mail: carolyn.hogg@sydney.edu.au

Preprint submitted at https://doi.org/10.1101/2022.02.13.480287

**Subjects** Animal and Plant Sciences, Genetics and Genomics, Ecology

## DATA DESCRIPTION

### Background and context

The numbat (*Myrmecobius fasciatus*; NCBI:txid55782) is a small marsupial (up to 700 g), and the only species within the Myrmecobiidae family [1] (Figure 1). Marsupials are one of three lineages of mammals, the others being eutherians (such as humans and mice) and monotremes (platypus, *Ornithorhynchus anatinus*; and echidna, *Tachyglossus aculeatus*) [2]. The Myrmecobiidae family is classified under the Dasyuromorphia order, which also contains carnivorous marsupials within the Dasyuridae family, such as the Tasmanian devil (*Sarcophilus harrisii*) and the extinct thylacine (*Thylacinus cynocephalus*, the family Thylacinidae) [3]. Numbats are unique amongst marsupials as they are the only diurnal marsupial. In addition, numbats have a variety of important adaptations to an arid environment, including regular torpor.

**Figure 1.** Adult numbat from the Australian Wildlife Conservancy. Photo credit: Wayne Lawler, Australian Wildlife Conservancy.

Numbats are the only termitivorous marsupial, consuming up to 20,000 termites per day [4]. Their high visual acuity, powerful front claws and sense of smell enables them to locate and dig out termite mounds and sub-surface structures [4]. Their extremely long tongues allow numbats to then extract termites from within the mounds. This bioturbation is important for ecosystem health, as numbats aerate the soil, facilitate seed germination and remove termite mud from hollows, thereby creating habitat for other species [5].

The numbat and other marsupials within the Dasyuromorphia, Notoryctemorphia and Diprotodontia orders undergo torpor [6]. Numbats are heterothermic endotherms, and regularly undergo shallow torpor to conserve energy during winter, characterised by a drop in body temperature for up to 15 h [7]. Marsupials typically have a low basal metabolic rate compared to eutherian mammals [8]. However, numbats' basal metabolic rate is even lower, at 82.5% that of other marsupials with an equivalent body mass [9, 10]. During torpor, metabolic rate can drop by up to 60% below the basal rate [6].

Historically, numbats inhabited the arid and semi-arid regions of Australia [11]. However, populations have declined by more than 99% due to habitat degradation and predation. It is estimated that there are only 1,000 individuals remaining in Western Australia [11]. As such, the numbat is currently listed as *endangered* ("EN") on the IUCN Red List [12] and *vulnerable* under the Australian Federal Government's Environment Protection and Biodiversity Conservation Act 1999. Numerous reintroductions to wild populations were conducted between 1985 and 2010; however, due to predation by introduced cats and foxes, many of these were failures, with only four being successful [13, 14]. Due to the ongoing threat of predation, numbats have also been released into several large-scale fenced enclosures to ensure the species' persistence [15]. The current numbat recovery plan recommends that additional subpopulations be established and the genetic health of all populations be maintained and measured [16].

Here we report the first *de novo* reference genome for the numbat, using 10× Genomics chromium linked-read sequencing. Assembly resulted in a 3.42 Gbp genome with a scaffold N50 of 223 k bp and 78.7% complete mammalian benchmarking universal single copy gene orthologs (BUSCOs v5.2.2) with 73.2% single-copy and 5.5% duplicated BUSCO genes [17]. A global transcriptome was also generated, consisting of transcripts from the liver, lung, and tongue. This was used to annotate the genome with Fgenesh++, resulting in 21,465 annotated genes with basic local alignment search tool (BLAST) hits to the National Centre for Biotechnology Information (NCBI) non-redundant database. Taste and vomeronasal receptors and aquaporin genes were manually annotated within the genome to investigate whether these gene families have expanded or contracted within the numbat, compared to other marsupials, in response to their unique life history. Annotation revealed a typically marsupial complement of vomeronasal and aquaporin gene families within the numbat, which does not reflect adaptation to their arid range. However, a subset of taste receptor genes have contracted in the numbat, compared to other marsupials, which may reflect adaptation to a termitivorous diet.

The numbat reference genome is a valuable tool for conservation and will be used alongside population genomic and genetic datasets to measure neutral and functional genetic diversity and the health of current and future populations. Obligate termitivorous mammals occur in multiple eutherian lineages, including the American anteaters, African aardvark, and the monotreme echidna. The numbat genome and transcriptomes generated in this study provide new insights into the molecular mechanisms which underpin the convergent evolution of this specialised dietary adaptation and the unique physiology of this iconic marsupial.

## METHODS
### Sample collection and sequencing
Numbat liver, lung and tongue were opportunistically sampled from a single female individual, housed at Perth Zoo, which in 2019 was euthanised due to medical reasons. All tissues were flash frozen at −80 °C and stored at this temperature until extraction. All samples were collected under Perth Zoo's opportunistic sampling standard operating procedure (export licence EF41000060) and with scientific licence number NSW DPIE SL101204.

High molecular weight (HMW) DNA was extracted from 25 mg of lung using the MagAttract HMW DNA kit (Qiagen) and quality was assessed using the NanoDrop 6000 with an A260/280 of 1.8 and A260/230 of 1.3. DNA was submitted to the Ramaciotti Centre for Genomics (UNSW) for 10× Genomics chromium library prep, and 150 bp paired-end (PE) reads were sequenced on an Illumina NovaSeq6000 S1 flowcell. This generated 143 GB of raw data, which was quality checked using F astQC v0.11.8 (RRID:SCR_014583) [18].

Total RNA was extracted from 25 mg of liver, lung and tongue, using the RNeasy Plus Mini Kit (Qiagen) with on-column DNA digestion using the RNase-free DNase I set (Qiagen). For the tongue, precise isolation of microscopic taste buds was difficult, but these structures are likely to have been included in the section of the tongue surface sampled. RNA purity was assessed using the NanoDrop 6000, with all samples displaying an A260/280 and A260/230 of 1.95 to 2.34. RNA concentration and integrity were measured using an RNA nano 6000 chip (Agilent Technologies), with all samples displaying an RNA integrity number (RIN) from 7 to 8.9. Total RNA was submitted to the Ramaciotti Centre for Genomics (UNSW)



for TruSeq mRNA library prep. All tissue libraries were sequenced as 150 bp PE reads across one lane of an S1 flow cell on the NovaSeq 6000. This resulted in 22–29 GB raw data per sample, which was quality checked using FastQC v0.11.8 (RRID:SCR_014583) [18].

## Genome assembly and annotation

*De novo* genome assembly was performed with Supernova v2.1.1 [19] using default parameters on Amazon Web Services (virtual machine 64 vCPUs; 976 GB RAM; 3 TB storage), obtaining approximately 64× raw coverage and 31× effective coverage. Assembly statistics were generated using BBTools (RRID:SCR_016968) [20], and assembly completeness assessed using BUSCO v5.2.2 and v3.1.0 (RRID:SCR_015008) [17]. The assembly was filtered to remove redundant haplotigs using SLiMSuite v1.8.1 [21]. Read representation was determined by trimming 10× adapters from the raw reads using BBmap (RRID:SCR_016965) [20], which were then mapped back to the assembly using BWA (RRID:SCR_010910) [22]. For annotation, a custom repeat database was generated for the genome using RepeatModeler v2.0.1 (RRID:SCR_015027) [23], then RepeatMasker v4.0.6 (RRID:SCR_012954) [24] was used to mask repeats, excluding low complexity regions and simple repeats. Genome annotation was then performed using Fgenesh++ v7.2.2 (RRID:SCR_018928) [25] with general mammalian pipeline parameters, and an optimised gene finding matrix from another species within the Dasyuromorphia order (Tasmanian devil: *Sarcophilus harrisii*). Transcripts with the longest open reading frame for each predicted gene were extracted from the global transcriptome and used as mRNA-based evidence for gene predictions. Similarly, the non-redundant metazoan protein database was used as protein-based evidence for gene predictions.

## Transcriptome assembly and annotation

Raw RNA sequencing (RNAseq) data was trimmed for quality and length using Trimmomatic v0.38 (RRID:SCR_011848) [26] with the following flags: ILLUMINACLIP:TruSeq3-PE.fa:2:30:10 SLIDINGWINDOW:4:5 LEADING:5 TRAILING:5 MINLEN:25. Illumina TruSeq sequencing adapters were removed from the dataset (ILLUMINACLIP:TruSeq3-PE.fa:2:30:10) as well as reads shorter than 25 bp (MINLEN:25). Reads were quality trimmed and removed where the average quality score fell below 5 within a 4 bp sliding window (SLIDINGWINDOW:4:5), as well as at the 5′ (LEADING:5) and 3′ (TRAILING:5) end of the read. Over 99.7% of reads were retained for all datasets, post trimming.

A global transcriptome for the numbat was generated *de novo* using trimmed reads from liver, lung and tongue as input to Trinity v2.8.3 (RRID:SCR_013048) [27] with default parameters and read normalisation. The Trinity script TrinityStats.pl was used to generate assembly statistics, representation of full-length protein-coding genes was determined by BLAST to Swiss-Prot, and completeness was assessed using BUSCO v5.2.2 and v3.1.0 (RRID:SCR_015008) [17] against the mammalian database. Functional annotation of the global assembly was performed using Trinotate v3.1.1 (RRID:SCR_018930) [28]. Briefly, TransDecoder v2.0.1 (RRID:SCR_017647) [27] was used to identify coding regions within transcripts, which in addition to the global assembly transcripts, were used to search: the Swiss-Protnon-redundant database, the Tasmanian devil reference genome annotations downloaded from NCBI (mSarHar1.11) and the immunome database of marsupials and monotremes [29] using BLAST+ [30] with an e-value cut-off of $1 \times 10^{-5}$ and HMMER v3.2.1 (RRID:SCR_005305). To determine the proportion of reads represented in the assembly,

trimmed reads were mapped back to the global transcriptome assembly using Bowtie2 v2.3.3.1 (RRID:SCR_016368) [31] with the flag -k 20 to indicate a maximum of 20 distinct alignments for each read. Alignments were used as input to transcript quantification with Salmon v1.4.0 [32] to generate transcripts per million (TPM) counts for each tissue. TransDecoder proteins expressed in the tongue with BLASTp hits to Swiss-prot (e-value of $10^{-5}$) were used as input to Panther (RRID:SCR_004869) [33] to assign gene ontology (GO) slim terms under the Biological Process and Molecular Function category.

## Manual gene annotation

Genes encoding taste and vomeronasal receptors, and aquaporins, were manually annotated in the numbat genome and transcriptomes to investigate gene expansion and/or contraction as a mechanism of evolutionary adaptation to the numbat's unique diet and historically arid range. Briefly, BLAST+ v2.7.1 [30] searches were conducted using known marsupial and eutherian sequences from each gene family as queries, with an e-value cut-off of 10 to ensure any potential hits were not excluded. Putative numbat genes for each family were aligned to other members from eutherians and marsupials using clustalW (RRID:SCR_017277) [34] in BioEdit (RRID:SCR_007361) [35] to confirm expected gene structure and presence of functional protein motifs. For TAS1R and vomeronasal receptor (VR) gene families the multiple sequence alignment was then used to construct phylogenetic trees in MEGAXv10.2.4 (RRID:SCR_000667) [36] for each family separately, using the neighbour-joining method with p-distance and 500 bootstrap replicates, as well as the maximum likelihood method with the James–Thornton–Taylor model. Both methods resulted in the same topology, so only the neighbour-joining trees are presented here. Phylogenetic analysis of TAS2R receptors was conducted in MEGAX v10.2.4 using the maximum likelihood method with the General Time Reversible model and gamma distribution with five categories. Bootstrap analysis was not performed as the topology of marsupial TAS2R sequences has been established previously [37]. VR genes were named in order of identification. Taste receptors and aquaporin genes were named according to clustering and bootstrap support within the phylogenetic tree. Extant marsupials have at least 27 orthologous gene groups (OGGs) of bitter taste receptor genes (TAS2Rs) [37]. TAS2R genes identified in the numbat genome were classified into marsupial OGGs based on their clustering with other known marsupial TAS2R genes within the phylogenetic tree in Figure 4. Accession numbers of sequences used as queries in BLAST+ and to generate phylogenetic trees are available in GigaDB [38].

## RESULTS AND DISCUSSION

### Genome

*De novo* assembly and subsequent filtering generated a 3.42 Gbp genome for the numbat with 30.97× coverage (Table 1). The genome contains 112,299 scaffolds with a scaffold N50 of 223 kb (Table 1) and is of a similar quality to the Tasmanian devil genome [39], but less contiguous than the antechinus (*Antechinus stuartii*) [40] and koala (*Phascolarctos cinerus*) genomes [37] (Table 1). The koala genome was generated using PacBio long-reads and multiple scaffolding technologies, so it is not surprising that this was more contiguous than the numbat assembly [37]. However, the antechinus genome was generated using 10× Genomics chromium linked-reads and assembled using the same Supernova v2.1.1 pipeline, yet was also more contiguous than the numbat genome [40] (Table 1). This difference in



**Table 1.** Numbat genome assembly statistics compared to koala, antechinus, Tasmanian devil, tammar wallaby and gray short-tailed opossum genomes, accessioned with the National Centre for Biotechnology Information (NCBI).

|  | Numbat | Koala | Antechinus | Tasmanian devil | Tammar wallaby | Gray short-tailed opossum |
|---|---|---|---|---|---|---|
| Reference | This study | [37] | [40] | [39] | [42] | [43] |
| Accession no. | – | GCF_002099425.1 | GCA_016696395.1 | GCF_000189315.1 | GCA_000004035.1 | GCF_000002295.2 |
| Year | 2021 | 2018 | 2020 | 2012 | 2011 | 2007 |
| Sequencing technology | 10× linked-reads | PacBio, Illumina, BioNano and HiC | 10× linked-reads | Illumina | Sanger ABI SOLiD | Whole-genome shotgun (WGS) method |
| Genome size (Gbp) | 3.42 | 3.19 | 3.31 | 3.17 | 3.07 | 3.59 |
| No. scaffolds | 112,299 | – | 30,876 | 35,974 | 277,711 | 5223 |
| No contigs | 219,447 | 1906 | 106,199 | 237,291 | 1,174,000 | 72,674 |
| Scaffold N50 (Mbp) | 0.223 | – | 72.7 | 1.8 | 0.0418 | 59.8 |
| Contig N50 (Mbp) | 0.037 | 11.58 | 0.08 | 0.02 | 0.0026 | 0.108 |
| GC (%) | 36.3 | 39.05 | 36.20 | 36.04 | 38.80 | 38.00 |
| Gaps (%) | 3.52 | 0.1 | 2.75 | 7.66 | 17.53 | 2.74 |

**Table 2.** Repeats elements annotated in the numbat genome. (Alu: Alu element; CR: complement receptor; ERV: endogenous retrovirus; LINE1: class 1 long interspersed nuclear elements; LINE2: class 2 long interspersed nuclear elements; LTR: long terminal repeat; MIRs: mammalian-wide interspersed repeats; SINE: short interspersed element.)

|  | Type of repeat | Number | Repeat sequence | |
|---|---|---|---|---|
|  |  |  | Length (bp) | Percent |
| SINE | Alu | 11,577 | 2,119,505 | 0.06% |
|  | MIRs | 1,718,103 | 243,123,222 | 7.10% |
| LINE | LINE1 | 1,402,806 | 733,512,048 | 21.42% |
|  | LINE2 | 807,962 | 176,657,028 | 5.16% |
|  | CR1 | 270,927 | 61,050,371 | 1.78% |
| LTR | ERVL | 2,082 | 762,885 | 0.02% |
|  | ERV1 | 21,263 | 6,393,956 | 0.19% |
|  | ERV2 | 16,540 | 5,447,399 | 0.16% |
| DNA elements | *hAT*-Charlie | 120,449 | 17,122,026 | 0.50% |
|  | *TcMar*-Tigger | 20,819 | 5,373,403 | 0.16% |
| Other | Unclassified | 1,034,362 | 236,046,652 | 6.89% |
|  | Small RNA | 641 | 57,442 | 0.00% |
|  | Satellites | 56,304 | 14,763,428 | 0.43% |

assembly contiguity may arise from molecule length, which represent reads with the same 10× barcode that align to the same region of a contig or scaffold [41]. This metric is an important contributing factor to the quality of 10× linked-read assemblies, with shorter molecules associated with reduced scaffold N50 and mis-assembly [41]. The antechinus genome had a molecule length of 74.08 kb [40], compared to only 23.13 kb for the numbat genome and below the recommended range of 50–100 kb by 10× Genomics [41]. HMW DNA of >40 kbp was used as input to sequencing in both species, although different extraction methods and kits were used in both cases, which may have contributed to the difference in molecule length [40]. In addition, the numbat HMW DNA may have degraded during transport, storage or sequencing, leading to fragmentation.

The repeat content of the numbat genome was similar to other marsupials, with 47.63% of the genome masked as repeats, compared to 44.82% in the antechinus [40] and 47.5% in the koala [37]. Several different repeat families were identified within the numbat genome (Table 2). Class I long interspersed nuclear elements (LINE1) and mammalian-wide interspersed repeats (MIRs) were the most numerous, as identified in other marsupials [37, 39, 40].

Annotation of the numbat genome with Fgenesh++ resulted in 77,806 gene predictions, of which 44,056 were supported by transcript evidence from the global transcriptome and



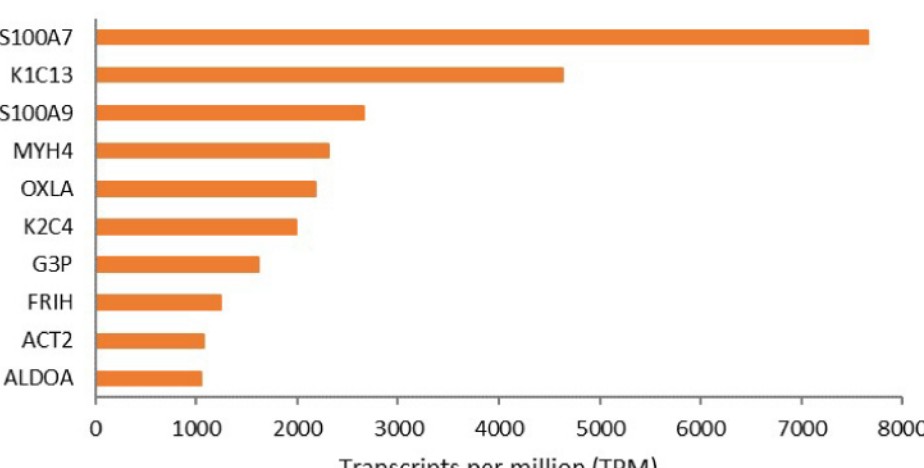

**Figure 2.** Top 10 transcripts expressed in the tongue with BLAST hits to Swiss-Prot proteins. Papillae were not discriminated at the time of tissue sampling. (ACT: actin; ALDOA: aldoase, fructose-biphosphate A; FRIH: ferritin heavy chain 1; G3P: glyceraldehyde-3-phosphate dehydrogenase; K1C13: keratin, type I cytoskeletal 13; K2C4: keratin, type II cytoskeletal 4; MYH4: myosin heavy chain 4; OXLA: OXA β-lactamases; S100: 100 Calcium Binding).

1,406 by protein evidence. Of these 77,806 genes, 21,465 had BLAST hits to eukaryote genes in the NCBI non-redundant database, which is similar to the number of annotated protein-coding genes in the Tasmanian devil (18,775) [39], antechinus (25,111) [40] and koala (26,558) [37].

## Transcriptome

The *de novo* global transcriptome, containing transcripts from the liver, lung and tongue, contained 2,119,791 transcripts, with an average length of 824 bp and transcript N50 of 1,393 bp. TransDecoder predicted 159,566 coding regions, of which 63% were complete (contained start and stop codons) and 86% had BLAST hits to Swiss-Prot. Given the numbat's specialised diet, we investigated the top 10 transcripts expressed within the tongue with BLAST hits to Swiss-Prot. Antimicrobial proteins from the S100 protein family and L-amino-acid oxidase [44] were highly expressed within the tongue (Figure 2). In addition, transcripts encoding keratin 13 and 4, which form part of the cytoskeleton, were also highly expressed, as well as myosin and actin proteins involved in muscle contraction (Figure 2). These transcripts reflect the structure and function of the tongue not only in feeding, but also as an epithelial barrier that forms a first line of defence against infection.

To further investigate adaptation to the numbat's unique life history, we manually annotated genes involved in taste, olfaction and water transport in the numbat genome. Duplication or pseudogenisation of genes within taste receptor, vomeronasal and aquaporin families have been identified in species with specialised diets. Bitter taste receptor and aquaporin genes have duplicated in the koala, likely due to their need to detoxify eucalypt leaves and ability to obtain water solely from their diet, without drinking [37]. Neofunctionalisation and pseudogenisation of taste receptor genes has also been linked to highly specialised diets. In large primates which eat a high proportion of leaves, including humans, the umami taste receptor gene has undergone

neofunctionalisation to recognise L-glutamate contained within leaves, which may promote leaf consumption [45]. In addition, bitter taste receptor genes which recognise leaf-derived toxins have also duplicated in some eutherian lineages [46]. On the other hand, in the giant panda (*Ailuropoda melanoleuca*), the umami taste receptor is not required for their herbivorous diet of bamboo; hence, the umami taste receptor gene is a pseudogene [47]. Our aim was to determine if similar duplication or pseudogenisation has occurred in the taste, vomeronasal and aquaporin gene families in the numbat genome in response to their termitivorous diet and semi-arid environment.

## Taste receptors

Two types of taste receptors are encoded within the genome and expressed within the mammalian oral cavity: type 1 (TAS1R) umami and sweet taste receptors, and type 2 (TAS2R) bitter taste receptors. Type 1 taste receptors are encoded by three genes in most mammals, including marsupials and monotremes: TAS1R1, TAS1R2 and TAS1R3 [37, 48]. The TAS1R1/TAS1R3 heterodimer and TAS1R2/TAS1R2 homodimer generate functioning umami and sweet taste receptors, respectively [49].

Orthologs of all three mammalian TAS1R genes were identified in the numbat genome (Figure 3, table in GigaDB [38]). Only partial sequences were characterised for TAS1R1 and TAS1R3, as short internal exons could not be identified. The TAS1R gene cluster was fragmented within the genome, as all three genes were encoded on different scaffolds compared to the single gene cluster in human and mouse [37, 49]. All TAS1R genes are likely functional in the numbat, owing to the lack of premature stop codons, insertions or deletions within the gene sequence, and identification of transcripts within the global transcriptome. Pseudogenisation of TAS1R genes has not occurred in the numbat in response to their specialised diet, as observed in the giant panda [47]. However, the expression of TAS1R1 and TAS1R3 in the numbat tongue transcriptome (TAS1R1 0.073573 TPM and TAS1R3 0.057388 TPM) may reflect their dietary preference for termites, as the TAS1R1/TAS1R3 heterodimer which forms the umami taste receptor recognises free nucleotides that are abundant in insects [45]. TAS1R2, which forms the sweet taste receptor, was not expressed in the tongue transcriptome, which may relate to the low level of free sugars in termites. While it is possible that TAS1R2 may be expressed in areas of the tongue not sampled, functional TAS1R2 genes are known to be expressed in non-taste organs for carbohydrate metabolism [50].

TAS2R are G-protein coupled receptors which detect bitter substances [49]. The number of TAS2R genes varies between species, with 26 and 40 genes in humans and mice, respectively [46]. Marsupials have several TAS2R genes which are orthologous to eutherians (TAS2R1, 2, 4, 38 and 60). However, some marsupials, such as the koala, have undergone large duplications within this gene family, with 66 TAS2R genes identified (24 intact and 42 pseudogenised) [37]. This expansion is thought to reflect adaptation to the koala's eucalypt diet, as duplications have occurred within TAS2R genes that presumably detect β-glucosides such as cyanogenic glycosides (TAS2R41 and 705), which are a component of eucalypt leaves [37]. Similarly, a large expansion of a sister OGG to eutherian and marsupial TAS2R41, TAS2R705 and TAS2R60, has also been identified in the echidna and platypus genome, indicating that this TAS2R gene cluster is highly conserved across mammals [48].

A total of 22 TASR2 genes were identified in the numbat genome, of which 11 were putative pseudogenes, owing to the presence of premature stop codons within the open



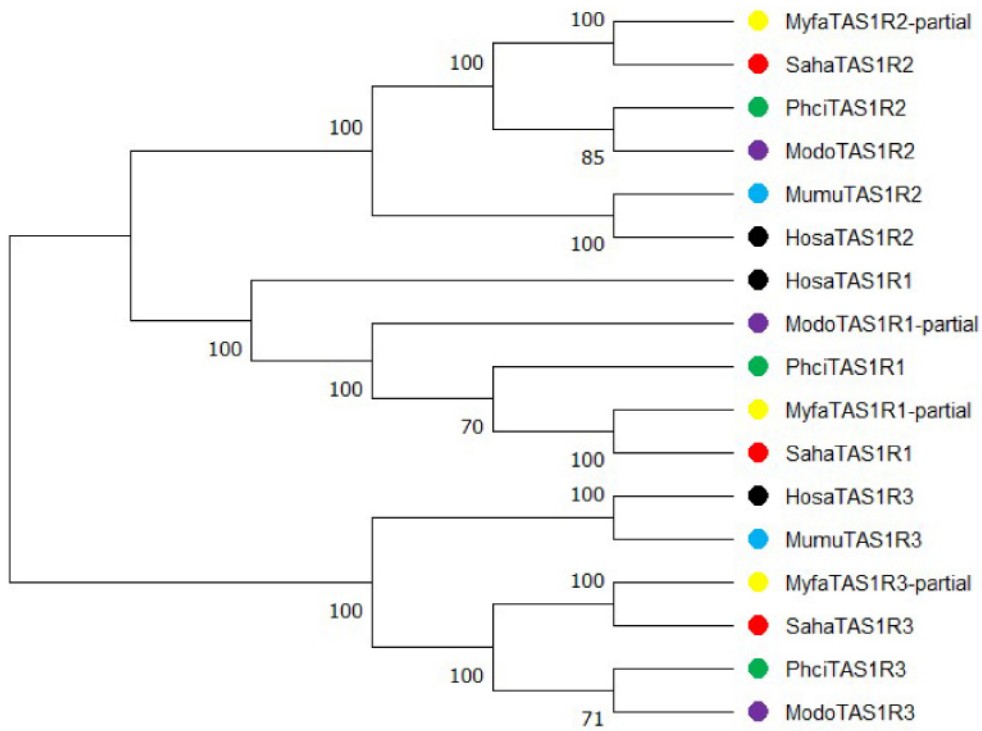

**Figure 3.** Phylogenetic relationship amongst numbat (Myfa; yellow), Tasmanian devil (Saha; red), koala (Phci; green), gray short-tailed opossum (*Monodelphis domestica*) (Modo; purple) and human (Hosa; black) TAS1R genes. The neighbour-joining phylogenetic tree was constructed using the p-distance method and 1,000 bootstrap values based on the amino acid sequence alignment. Only bootstrap values greater than 50% are shown.

reading frame (Figures 4, 5; table in GigaDB [38]). While only two of the 11 numbat TAS2R genes with complete coding sequences were identified in the global transcriptome, this is likely due to the fact that TAS2Rs are mainly expressed within papillae, which were not discriminated at the time of tongue tissue sampling [49]. The number of TAS2R genes in the numbat was the smallest amongst marsupials studied to date, and more similar in size to monotremes [37] (Figure 4). Despite this, numbat TAS2R genes cluster within the OGG clade containing other marsupial and eutherian TAS2R genes which may detect harmful β-glucosides (TAS2R41, 60 and 705) contained within arthropods (Figure 5). The insectivorous echidna (*Tachyglossus aculeatus*) has a similarly reduced TAS2R gene repertoire, compared to the platypus. However, the OGG of TAS2R41, 60 and 705 have also been retained in the echidna, as observed in the numbat [48]. The presence of this OGG and an overall reduction of the TAS2R gene repertoire in these two species may reflect adaptation to an insectivorous diet. However, TAS2R gene contraction in the numbat may also be due to assembly error. Numbat TAS2R genes were encoded across six scaffolds, compared to two main gene clusters in the human and mouse genome [49]. Future improvements to numbat genome contiguity may uncover additional taste receptor genes in the numbat, enabling reconstruction of gene clusters and investigation of synteny within this genomic region.

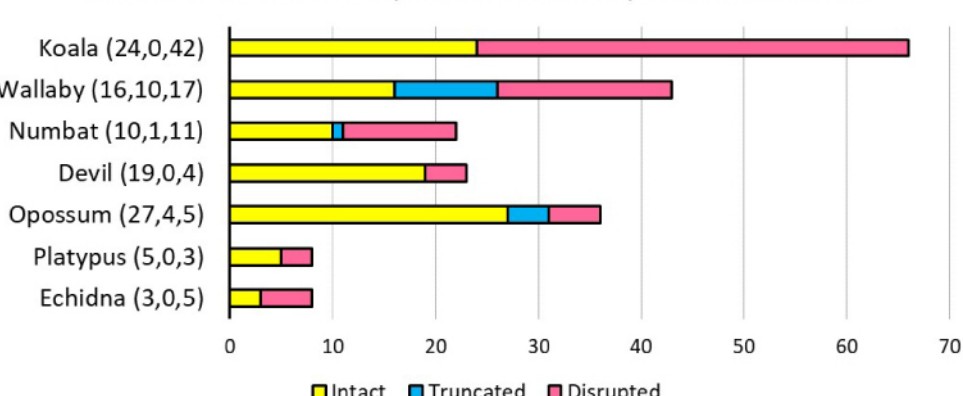

**Figure 4.** Number of type 2 taste receptor (TAS2R) genes in the numbat compared to other marsupials and monotremes with intact coding sequences (CDS), truncated CDS or disrupted CDS which likely represent pseudogenes. TAS2Rs with truncated CDS may represent incomplete sequences due to short contigs and/or scaffolds, or pseudogenes.

## Vomeronasal receptors

VRs are a class of olfactory receptors primarily expressed in the vomeronasal organ (VNO) within the nose and involved in the detection of pheromones. There are two types of VRs, type 1, (V1R) and type 2 (V2R), encoded by separate gene families which differ in their expression pattern and gene structure [51]. Binding of pheromones or odorants to V1R and V2R initiates chemical sensing that has important roles in many behaviours, such as mating and aggression. The number of genes encoding V1R and V2R differs amongst species, with many mammals displaying a discrepancy in the ratio of V1R to V2R genes [51].

V1Rs are involved in the detection of small pheromones within the air, such as those involved in sex and mating [51, 52]. V1Rs are encoded by an intronless gene and are primarily expressed within the apical layer of the epithelium within the VNO [51]. The majority of V1Rs in humans are pseudogenes [53], while rodents have a large expansion of more than 200 functional genes [54]. A total of 162 V1R genes were identified in the numbat genome, of which 112 contained intact coding sequences and 5 were expressed in the global transcriptome (table in GigaDB [38]). The lack of expression for the majority of numbat V1Rs is not surprising given that they are solely expressed within the VNO in other mammals [51]. The 112 putative functional numbat V1R genes were encoded across 68 different scaffolds. This is compared to large clusters of duplicated genes in the mouse genome, indicating the VR gene family was highly fragmented in the numbat genome, similar to taste receptors.

The numbat V1R gene repertoire is similar to other marsupials and monotremes, with more than 90 genes identified in the opossum [52] and tammar wallaby [52, 54] and 280 genes in the platypus [54, 55]. V1Rs from the numbat and other marsupials form both marsupial-specific and species-specific clades in the phylogenetic tree, which may reflect marsupial-specific adaptations (Figure 6). For example, VRs are thought to be involved in the unaided movement of altricial marsupial young from the birth canal to the mother's pouch and teat [56].

V2Rs are expressed in the basal layer of the VNO epithelium and detect water-soluble peptides and pheromones [51, 52]. Similar to V1Rs, V2R gene number varies significantly

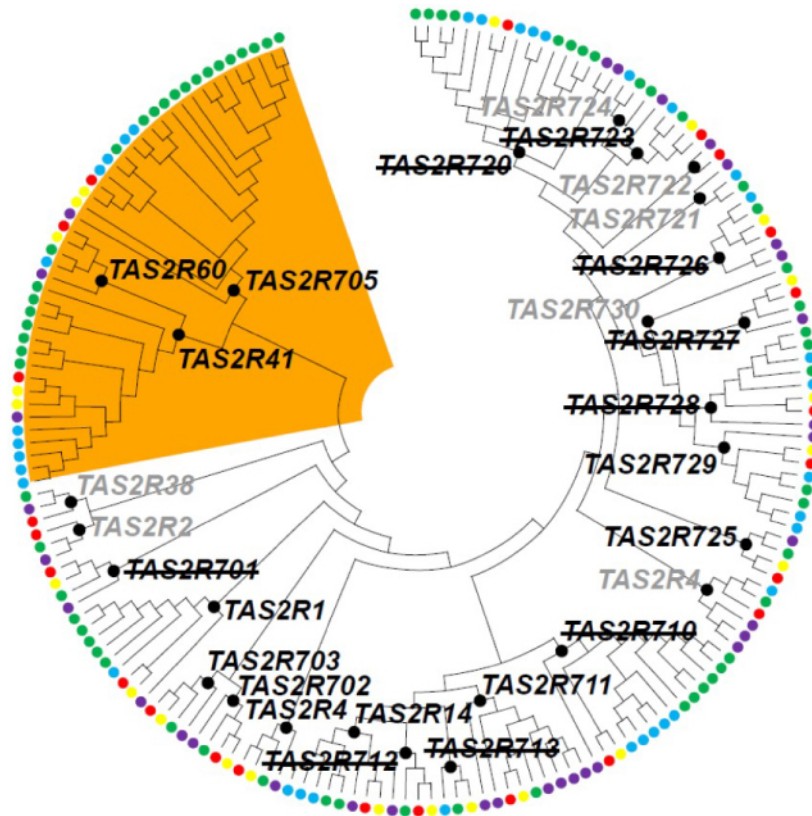

**Figure 5.** Phylogenetic relationship amongst marsupial type 2 taste receptor (TAS2R) genes. Numbat TAS2R genes are indicated by the yellow circle and the other circle colours mirror Figure 3 with the addition of tammar wallaby (*Notamacropus eugenii*) indicated by the blue circle. TAS2R sequences of non-numbat marsupials were annotated in [37]. The phylogenetic tree was constructed using the Maximum Likelihood method (GTL+G, 5 categories) based on the nucleotide sequence alignment. Only topology is shown. The clades of 27 marsupial OGGs are indicated by the black circles around each respective node with the corresponding gene name. TAS2R gene names with a strikethrough (e.g., TAS2R701) indicate the corresponding numbat TAS2R gene is pseudogenised, and grey gene names indicate the corresponding numbat TAS2R gene is absent. The marsupial OGGs which may recognise β-glucoside are shaded in orange, note the numbat TAS2R genes within this clade (TAS2R60, 705 and 41) are intact within the genome.

amongst species [51]. V2R genes have expanded in rodents, with more than 100 and functional genes and 150 pseudogenes [57]. In comparison, the platypus genome contains 15 V2Rs [57, 58], while humans and primates do not encode functional V2R genes [52, 53, 57].

A total of 29 V2R sequences were identified in the numbat genome, of which 22 likely represent functional genes (table in GigaDB [38]). The number of V2R genes in the numbat is low compared to the 86 functional (70 pseudogene) V2R genes in opossums [52]. The low number of V2R genes identified in the numbat may be due to assembly fragmentation, as V2R genes were encoded across 17 scaffolds, many of which were short and only contained partial V2R sequences. However, V2Rs have not been manually annotated in other marsupial genomes, which hinders our interpretation of these results.

Overall, numbats encode a number of functional type 1 and 2 VRs, unlike many eutherian mammals. While V1R genes have expanded in numbats, as in other marsupials,

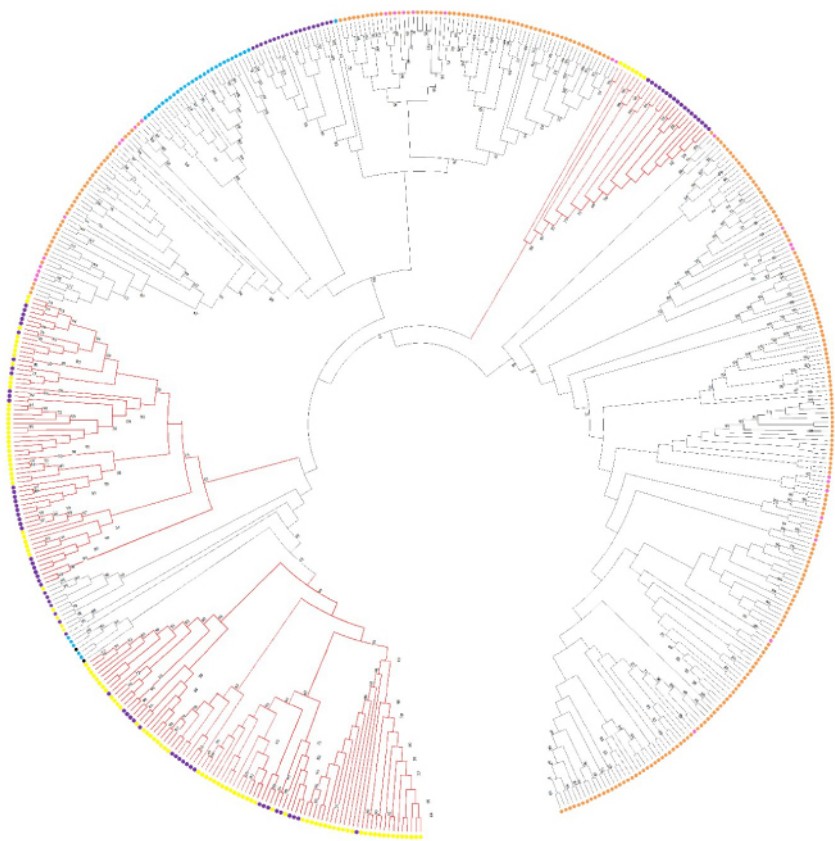

**Figure 6.** Phylogenetic relationship amongst numbat (yellow), opossum (purple), human (black), mouse (blue), platypus (orange) and echidna (pink) type 1 vomeronasal receptor genes. Marsupial-specific clades are denoted by the red branches. The neighbour-joining phylogenetic tree was constructed using the p-distance method and 1,000 bootstrap values based on the amino acid sequence alignment. Only bootstrap values greater than 50% are shown.

the lack of annotated marsupial V2R genes limits our ability to identify if the low gene number in numbat results from incomplete gene annotations or gene family contraction.

## Aquaporins

Numbats historically inhabited the arid and semi-arid areas of southern and central Australia [11]. Unlike many arid marsupials, the numbat's renal morphology and urinary concentration does not reflect this environment [59]. However, gene families involved in water metabolism, such as aquaporins, have not been explored. Aquaporins are plasma membrane channels involved in the transport of water and other small molecules, and are essential for water balance [60]. Aquaporin genes have undergone duplications in the koala, an adaptation to their highly specialised diet of eucalypt leaves and ability to "taste water" [37].

A total of 12 aquaporin genes were identified in the numbat genome, including the water-selective aquaporins (AQP1, 2, 4, 5, 6 and 8), aquaglyceroporins (AQP3, 7, 9 and 10) and superaquaporins (AQP11 and 12) [60] (table in GigaDB [38]). This gene content is identical to most marsupials [37] and similar to the 13 aquaporin genes in humans [60]. All 12 numbat aquaporins were expressed in the global transcriptome, and all but AQP2 were

expressed in the tongue. AQP5 is highly expressed in the tongue of other mammals as it is central to sensing water concentration [60]. This was not the case for the numbat, as AQP5 was expressed in the tongue at low levels (0.013 TPM) whereas AQP3 was the most highly expressed aquaporin in this tissue (62.9 TPM). The numbat aquaporin gene repertoire is typical of other mammals and has not undergone gene duplications as observed in the koala [37].

### DATA VALIDATION AND QUALITY CONTROL

Functional completeness of the genome and global transcriptome was assessed by searching for the presence of single copy gene orthologs using BUSCO [17]; We identified 78.7% and 76.4% of complete mammalian BUSCO v5.2.2 genes, and 82.7% and 72% of complete mammalian BUSCO v3.1.0 genes, in the genome and global transcriptome, respectively. These BUSCO scores are lower than reported for the koala [37], Tasmanian devil [39] and antechinus genomes [40], indicating the draft status of the numbat genome. Despite this, 93.88% of input reads mapped to the genome assembly, and more than 96% of trimmed RNAseq reads from each of the three tissues mapped to the global transcriptome. This indicates that the genome and transcriptome assemblies are an accurate representation of the input sequencing reads.

### REUSE POTENTIAL

The *de novo* assembly of the numbat genome using 10× Genomics chromium linked-reads resulted in a 3.42 Gbp draft-quality genome. As the numbat is the sole member of the Myrmecobiidae family and the only diurnal and termitivorous marsupial, this genome provides an opportunity to study the genetic basis of these unique traits. The numbat genome is one of few arid marsupial genomes that have been sequenced, and represents an important contribution to studying adaptation to aridity, particularly given climate change.

The numbat genome can immediately be used for conservation management through alignment of population genetics datasets, such as reduced representation sequencing, which will enable monitoring of both genome-wide and functional genetic health and diversity of numbat populations. Despite the fragmented nature of the genome, the draft numbat assembly has enabled investigation of taste, vomeronasal and aquaporin gene families in this unique marsupial and provides a basis for future sequencing projects.

### CONCLUSION

We have generated a draft genome assembly and global transcriptome assembly for the numbat, the only member of the Myrmecobiidae family and the only termitivorous marsupial. Given the numbat's specialised diet, we investigated highly expressed transcripts within the tongue, and manually annotated taste and vomeronasal receptors in the genome. The tongue contains numerous transcripts involved in feeding and immunity, highlighting its role as a first line of defence against foreign agents. The pattern of taste receptor expression in the tongue and putative contraction of the bitter taste receptor gene repertoire in the numbat genome may reflect their specialised termitivorous diet. VR gene families in the numbat did not show evidence of gene expansion or contraction as observed in other mammals with specialised diets. Similarly, numbat aquaporin genes were similar to other mammals and did not reflect adaptation to an arid environment. However, genome fragmentation influenced the quality of manual gene annotation and further work



is required for confirmation. The numbat genome is an important resource for conserving this distinctive marsupial and understanding its unique life history and termitivorous diet.

## DATA AVAILABILITY

The numbat genome and global transcriptome assembly supporting the results of this article are available through Amazon Web Services open datasets program https://registry.opendata.aws/australasian-genomics/. The genome assembly and all raw sequencing reads, including the 10× linked-reads and RNAseq reads, are available through NCBI under BioProject number PRJNA786364. Data is also available in the *GigaScience* GigaDB repository [38].

## DECLARATIONS
## LIST OF ABBREVIATIONS

AQP: aquaporin; BLAST: basic local alignment search tool; bp: base pair; BUSCO: benchmarking universal single copy orthologs; CDS: coding sequences; HMW: high molecular weight; kbp: kilobase pair; LINE1: class 1 long interspersed nuclear elements; Mbp: megabase pair; MIRs: mammalian-wide interspersed repeats; NCBI: National Centre for Biotechnology Information; OGGs: orthologous gene groups; PE: paired-end; RNAseq: RNA sequencing; TAS1R: type 1 taste receptor; TAS2R: type 2 taste receptor; TPM: transcripts per million; V1R: type 1 vomeronasal receptor; V2R: type 2 vomeronasal receptor; VNO: vomeronasal organ; VR: vomeronasal receptor.

## ETHICS APPROVAL AND CONSENT TO PARTICIPATE

Not applicable.

## COMPETING INTERESTS

The authors declare that they have no competing interests.

## AUTHORS' CONTRIBUTIONS

PB and LS assembled and annotated the genome. EP assembled and annotated the global transcriptome, conducted transcript counts and manually annotated taste and vomeronasal receptors, and aquaporin genes. TH analysed molecular evolution of taste receptors. KB and CJH designed the study. All authors viewed, commented on and agreed to publication of the manuscript.

## FUNDING

This work has been funded by the Australian Research Council Centre of Excellence for Innovations in Peptide and Protein Science (CE200100012) and Discovery Project (DP180102465), and the Bioplatforms Australia Oz Mammals Genomics consortia. LS and PB are each supported by an Australian postgraduate award. TH is supported by a KAKENHI grant from the Japan Society for the Promotion of Science (JSPS) (19K16241, 21H04919, and 21KK0106) and the JSPS Bilateral Joint Research Project "Earth-wide comparative genomics of endangered mammals in Japan and Australia" (JPJSBP 120219902).

## ACKNOWLEDGEMENT

The authors would like to acknowledge Perth Zoo for providing samples.

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
