## [Reviewer Report]

Comments on revised manuscriptI thank the authors for addressing the comments I previously provided and generally find them satisfactory. I have listed a few small typos below that I found and add a general comment on marsupial diurnality for the author’s consideration (though do not think that it necessitates further amendments to the text). 

I recommend the article for acceptance. Thank you for the opportunity to review it.

Warm Regards,
Charles

Minor comments: 
-Line 16: “Critically endangered” is an official IUCN conservation status (e.g. the Woylie is critically endangered). As you point out later, IUCN lists numbats as “endangered”. To avoid confusion with the wrong conservation status, I would change this to just say “endangered”

-Line 19: Chromium should be capitalized

-Citation 13: Lists the Woylie’s IUCN report rather than numbat. 
Swap:
[1] Woinarski J and Burbidge AA: Bettongia penicillata The IUCN Red List of Threatened Species 2016: e. T27858A21961347. https://dx.doi.org/10.2305/IUCN.UK.2016-2.RLTS.T2785A21961347.en (2016). Accessed 25th November 2021.
With:
[2] Woinarski, J. & Burbidge, A.A. 2016. Myrmecobius fasciatus. The IUCN Red List of Threatened Species 2016: e.T14222A21949380. https://dx.doi.org/10.2305/IUCN.UK.2016-2.RLTS.T14222A21949380.en. Accessed on 15 February 2022.)

-Lines 45-46 “Numbats are unique amongst marsupials as they are the only diurnal and marsupial.” There is an extra "and" between diurnal and marsupial. 

-Comment regarding marsupial diurnality: Unrelated to this review, I was directed a while ago by a colleague to Torodov et al. 2021 (https://doi.org/10.1098/rspb.2021.0394) for a database of marsupial trait data. I happened to look back at this and in their supplements they list 5 other extant marsupial species as being diurnal (the broad-striped dasyure, western brush wallaby, whiptail wallaby, emilia’s short-tailed opossum, and shrewish short-tail opossum). That said, the literature on these species seems very sparse. I wouldn't insist that the authors change their comments on the numbat being uniquely diurnal among marsupials on the basis of a single paper and its particular choices about where the line between 'diurnal' and 'diurnal-crepuscular' lies. However, I do think the numbat’s singular status as a diurnal marsupial is a bit of a grey area.

---

## [Reviewer Report]

Upload additional filesDRR-202112-02/form/Numbat Genome Reviewer Comments - CYF (2).docxReviewer name and names of any other individual's who aided in reviewer Charles Feigin (reviewer), Elise Ireland (assisted in proofreading)Do you understand and agree to our policy of having open and named reviews, and having your review included with the published papers. (If no, please inform the editor that you cannot review this manuscript.)YesIs the language of sufficient quality?YesPlease add additional comments on language quality to clarify if needed
Are all data available and do they match the descriptions in the paper? YesAdditional CommentsI selected "yes" to the question "Are all data available and do they match the descriptions in the paper?" as the genome, annotation, transcriptome etc are currently available through the author's institutional/consortium AWS link. However, so far as I can tell the BioProject number they provide is not yet public. This would contain the same data, but a release in the public repository is a more secure way ensure data is permanently accessible to the community.Are the data and metadata consistent with relevant minimum information or reporting standards? See GigaDB checklists for examples <a href="http://gigadb.org/site/guide" target="_blank">http://gigadb.org/site/guide</a>YesAdditional CommentsIs the data acquisition clear, complete and methodologically sound?YesAdditional CommentsIs there sufficient detail in the methods and data-processing steps to allow reproduction?NoAdditional CommentsI have requested in my comments for a few program version numbers, parameters, sequence accession numbers to be included and suggested a few points of clarification on the methods. This should be very straightforward for the authors to address.Is there sufficient data validation and statistical analyses of data quality? YesAdditional CommentsIs the validation suitable for this type of data?YesAdditional CommentsIs there sufficient information for others to reuse this dataset or integrate it with other data?YesAdditional CommentsAny Additional Overall Comments to the AuthorRecommendationMinor Revision

---

## [Reviewer Report]

Reviewer name and names of any other individual's who aided in reviewer Xu WangDo you understand and agree to our policy of having open and named reviews, and having your review included with the published papers. (If no, please inform the editor that you cannot review this manuscript.)YesIs the language of sufficient quality?YesPlease add additional comments on language quality to clarify if needed
Are all data available and do they match the descriptions in the paper? NoAdditional CommentsPRJNA786364 cannot be found at NCBI. The numbat assembly can be assessed at AWS https://threatenedspecies.s3.ap-southeast-2.amazonaws.com/index.html#Myrecobius_fasciatus/. Are the data and metadata consistent with relevant minimum information or reporting standards? See GigaDB checklists for examples <a href="http://gigadb.org/site/guide" target="_blank">http://gigadb.org/site/guide</a>YesAdditional CommentsIs the data acquisition clear, complete and methodologically sound?YesAdditional CommentsIn this manuscript, Peel et al. sequenced and assembled a draft genome of Myrmecobius fasciatus, a termitivorous marsupial species known as the numbat. A total of 94% sequencing reads could be mapped to the draft assembly, and 96% of RNA-seq reads from three tissues can be aligned. This is the first genome assembly for this species and provides the necessary genome resource for molecular study of the interesting characteristics in this species. Is there sufficient detail in the methods and data-processing steps to allow reproduction?NoAdditional CommentsOne major concern is the completeness of this assembly. The BUSCO analysis revealed a completeness score of 83%, missing one-fifth of mammalian BUSCO genes. The missing gene models in this assembly may affect the inference of gene gain/loss and gene family expansion. The authors should discuss this, so the audience can be aware of this when using this assembly for comparative genomic analysis. 

Please find the specific comments below: 
1. To determine whether this assembly is incomplete or not, a genome size estimation analysis should be performed based on K-mers or other approaches. 
2. How many chromosomes are their in numbat genome? Could the authors describe the katyotype if it has been characterized in the previous literature? 
3. Line 73, 82.8% complete BUSCO: could the authors break this down to single-copy, multiple-copy BUSCOs? 
4. Line 95, 10x Genomics does not support genomes larger than the human genome. How much input DNA the authors used for the 10x library prep? 
5. Line 190, Table 1: please add the Monodelphis and Tammy wallaby genome assembly statistics, which are the first two assembled marsupial genomes. 
6. Line 193, the numbat repeat content (47.6%) was compared to antechinus (44.8%) and koala (47.5%). I suggest that the authors add the two species in Table 2, to check if any specific classes of repetitive elements were enriched in numbat. 
7. Did the authors assess the level of potential contamination from microbial species? 
8. I suggest the authors talk the top 20 largest scaffolds, and perform a syntenic analysis compared to tammar wallaby or the koala genome. 
9. Line 295. Vomeronasal receptors: marsupials have a large number of olfactory receptors. Have the authors check the total number of olfactory receptor genes in the numbat, and how does it compare to other marsupial species? 
Is there sufficient data validation and statistical analyses of data quality? YesAdditional CommentsIs the validation suitable for this type of data?YesAdditional CommentsIs there sufficient information for others to reuse this dataset or integrate it with other data?NoAdditional CommentsPRJNA786364 cannot be found at NCBI. Please release the dataset after acceptance. Any Additional Overall Comments to the AuthorRecommendationMinor Revision